# *MYH11* rare variant augments aortic growth and induces cardiac hypertrophy and heart failure with pressure overload

Zhen Zhou[1], Kgosi Hughes[1], Nisha Saif[1,2], Hyoseon Kim[3], Michael P. Massett[3], Mingjie Zheng[4], Alana C. Cecchi[1], Dongchuan Guo[1], David R. Murdock[1], Ping Pan[1], Jelita S. Clinton[1], Jun Wang[4], John M. Greally[5], Dianna M. Milewicz[1]*

1 Division of Medical Genetics, Department of Internal Medicine, The University of Texas Health Science Center at Houston McGovern Medical School, Houston, Texas, United States of America, 2 Broad Institute of MIT and Harvard, Cambridge, Massachusetts, United States of America, 3 Department Kinesiology & Sport Management, Texas Tech University, Lubbock, Texas, United States of America, 4 Department of Pediatrics, The University of Texas Health Science Center at Houston McGovern Medical School, Houston, Texas, United States of America, 5 Department of Genetics, Albert Einstein College of Medicine, New York, New York, United States of America

* Dianna.M.Milewicz@uth.tmc.edu

## Abstract

Smooth muscle cell-specific myosin heavy chain, encoded by *MYH11*, is selectively expressed in smooth muscle cells (**SMC**s). Pathogenic variants in *MYH11* predispose to a number of disorders, including heritable thoracic aortic disease associated with patent ductus arteriosus, visceral myopathy, and megacystis-microcolon-intestinal hypoperistalsis syndrome. Rare variants of uncertain significance occur throughout the gene, including *MYH11* p.Glu1892Asp, and we sought to determine if this variant causes thoracic aortic disease in mice. Genomic editing was used to generate *Myh11*E1892D/E1892D mice. Wild-type (**WT**) and mutant mice underwent cardiovascular phenotyping with and without transverse aortic constriction (**TAC**). *Myh11*E1892D/E1892D and WT mice displayed similar growth, blood pressure, root and ascending aortic diameters, and cardiac function up to 13 months of age, along with similar contraction and relaxation on myographic testing. The hypertension induced by TAC was similarly in *Myh11*E1892D/E1892D and WT mice, but mutant mice showed augmented ascending aortic enlargement and increased elastic fiber fragmentation on histology. Unexpectedly, male *Myh11*E1892D/E1892D mice undergoing TAC had decreased ejection fraction, stroke volume, fractional shortening, and cardiac output compared to similarly treated male WT mice. Importantly, left ventricular mass increased significantly due to primarily posterior wall thickening, and cardiac histology confirmed cardiomyocyte hypertrophy and increased collagen deposition in the myocardium and surrounding arteries. These results further highlight the phenotypic heterogeneity associated with *MYH11* rare variants. Given that *MYH11* is selectively expressed in SMCs, these results implicate a role of SMCs in the arteries of the heart contributing to cardiac hypertrophy and failure with pressure overload.

**Data availability statement:** All data supporting the findings of this study are available within the main text and the Supplementary Information.

**Funding:** This research was funded by the Leducq Foundation (22CVD03), the National Heart, Lung, and Blood Institute (1P01 HL169168-01A1, R01 HL109942, and 5R01 HL146583), and the John Ritter Foundation to DMM. The funders had no role in study design, data collection and analysis, decision to publish, or preparation of the manuscript.

**Competing interests:** The authors have declared that no competing interests exist.

## Author summary

In this study, we explore the impact of a specific genetic variant, *MYH11* p.Glu-1892Asp, on the heart and blood vessels in mice. The *MYH11* gene is crucial for smooth muscle cells, which are found in the walls of blood vessels and play an important role in various vascular diseases. We created mice with this genetic variant to see if it would lead to thoracic aortic disease, a condition affecting the main artery from the heart. We found that mice with the variant were similar to normal mice in many aspects, such as growth, blood pressure, and heart function, up to 13 months of age. However, when we induced high blood pressure in the mice, the mutant mice showed more significant enlargement of the aorta and damage to the elastic fibers in the aortic walls. Interestingly, male mutant mice also developed heart problems, such as reduced heart pumping ability and increased heart muscle thickness, when high blood pressure was induced. This was accompanied by heart muscle cell enlargement and increased fibrosis of the cardiac tissues. These findings suggest that this rare *MYH11* variant can contribute to a range of heart and vascular issues, particularly under conditions of pressure overload, and highlight the importance of smooth muscle cells in the development of these cardiovascular complications.

## Introduction

*MYH11* encodes the smooth muscle-specific isoform of myosin heavy chain (**SMMHC**), which associates with a regulatory light chain and a second light chain of unknown function to form the thick filament in the contractile unit of smooth muscle cells (**SMC**s) [1]. *MYH11* is highly and selectively expressed in SMCs, as illustrated by the fact that endogenous *Myh11* expression is restricted to SMCs in *Myh11* knock-in reporter mice [2–4], the *Myh11* promoter is used as a Cre-driver to lineage-trace SMCs [5–9], and single-cell transcriptomic analyses in various tissues and developmental stages demonstrate that *Myh11* expression is limited to SMCs [10,11]. Pathogenic variants in *MYH11* confer a highly penetrant risk for several disorders, including heritable thoracic aortic disease associated with patent ductus arteriosus [12–16]. Although rare missense variants are present throughout *MYH11*, the majority of pathogenic variants that cause heritable thoracic aortic disease are large, in-frame deletions in the coiled-coil domain, a region that is critical for polymerization of SMMHC into thick filaments. Rare chromosomal duplications of 16p13.1 that include *MYH11* and eight other genes also confer an increased risk for aortic dissection. However, there is no evidence that the corresponding deletion increases the risk for thoracic aortic disease (**TAD**) [17]. Instead, recessive loss-of-function *MYH11* pathogenic variants are responsible for fetal megacystis-microcolon. Finally, heterozygous variants that disrupt the termination codon at the C-terminus and add additional amino acids to the end of the protein predispose individuals to a smooth muscle dysmotility syndrome with esophageal, gastric, and intestinal complications [18–20].

*MYH11* variants of uncertain significance (**VUS**s) are commonly reported in genetic testing, but the phenotypic variability and burden of *MYH11* rare variants make it difficult to assign pathogenicity to identified variants. We previously determined that a VUS in *MYH11*, p.Arg247Cys (R247C), decreases myosin motor function in *in vitro* assays [21]. *Myh11*[R247C/R247C] mice show decreased aortic ring contraction, yet have normal growth, survival, and no evidence of TAD [21]. However, when hypertension is induced using 3g/L L-N[G]-Nitro arginine methyl ester and a high-salt diet, one-fifth of *Myh11*[R247C/R247C] mice die due to acute dissection of the proximal aorta [22]. Additionally, a heterozygous in-frame deletion in *MYH11*, p.Lys1256del, was identified to segregate with thoracic aortic dissection in two independent pedigrees [14]. Mice that are homozygous for the *MYH11* p.Lys1256del alteration also had no aortic disease up to 18 months of age, but aortic dissections occur at a higher rate in both heterozygous and homozygous mutant mice compared to wild-type (**WT**) mice with angiotensin II infusion [23]. These data support that rare variants in *MYH11* can contribute to increased risk for TAD.

*MYH11* missense VUSs in the coiled-coil region have not been functionally assessed in mice, so we sought to determine if *MYH11*, p.Glu1892Asp (E1892D) increases the risk for TAD in a mouse model. *Myh11*[E1892D/E1892D] mice did not develop TAD with age, but when male *Myh11*[E1892D/E1892D] mice were subjected to transverse aortic constriction (**TAC**) for 2 weeks, augmented ascending aortic enlargement occurred compared to WT mice. Unexpectedly, TAC also induced significant cardiomyocyte hypertrophy, increased cardiac fibrosis, and impaired left ventricular contractile function in the *Myh11*[E1892D/E1892D] mice. These studies broaden our understanding of the phenotypes associated with *MYH11* rare variants and identify a novel role for mutant arterial SMCs contributing to aberrant cardiac remodeling with pressure overload in mice.

## Results

### VUS in *MYH11*, p.Glu1892Asp, identified in a patient with TAD

The proband is a 66-year-old male of European descent with an aortic root that progressively increased from 4.2 to 4.8 cm over 10 years. He underwent a successful valve-sparing aortic root and ascending aorta repair. The proband has pectus excavatum, pes planus, and myopia, but no other skeletal, cardiac, or ocular abnormalities. He also has hypercholesterolemia and hypertension that are controlled with medications. His father had a 4.2 cm aortic root and died of lung cancer at age 68, and his paternal grandfather died suddenly of an unknown cause at the age of 50 years. There was no other family history of TAD or sudden death.

Genome sequencing performed on DNA isolated from both peripheral blood leukocytes and tissues from the resected aorta showed no evidence of pathogenic variants in known aortopathy genes, but revealed a VUS in *MYH11*, p.Glu-1892Asp (c.5676G > C; CADD score 23.70 and REVEL score 0.562). The variant is located in the α-helical coiled-coil domain at the C-terminus of the protein and occurs in the gnomAD database at a frequency of ~0.6% in European populations, ~0.2% in South Asian and African/African-American populations, and is not found in East Asian populations. The relatively high frequency of the variant in some populations led to its categorization as benign or likely benign in ClinVar (VCV000138358.34).

### Validation and cardiovascular phenotyping of *Myh11*[E1892D/E1892D] mice

The *MYH11* p.E1892D variant was introduced into the mouse genome using CRISPR/Cas editing of C57BL/6J embryos. Sequencing of mouse tail DNA and cDNA from both heart and thoracic aortic tissues confirmed that the *Myh11* variant was present in genomic DNA and the expressed transcript ([Fig 1A]). A synonymous missense variant, p.Ser1893Ser (c.5679C > A), was introduced to prevent re-cutting during genome editing, had no impact on mRNA splicing based on SpliceAI analysis (score 0). A total of 102 progenies from heterozygous breeders were screened, and expected Mendelian ratios of the variant were obtained ([S1 Table]).

*Myh11*[E1892D/E1892D] mice and littermate controls (10 males and 10 females) underwent cardiovascular phenotyping every 6 weeks up to 13 months of age. *Myh11*[E1892D/E1892D] mice grew normally, maintained a blood pressure similar to WT mice, and had a normal heart rate. Cardiovascular assessment found that blood pressure and growth of the root and ascending

PLOS Genetics

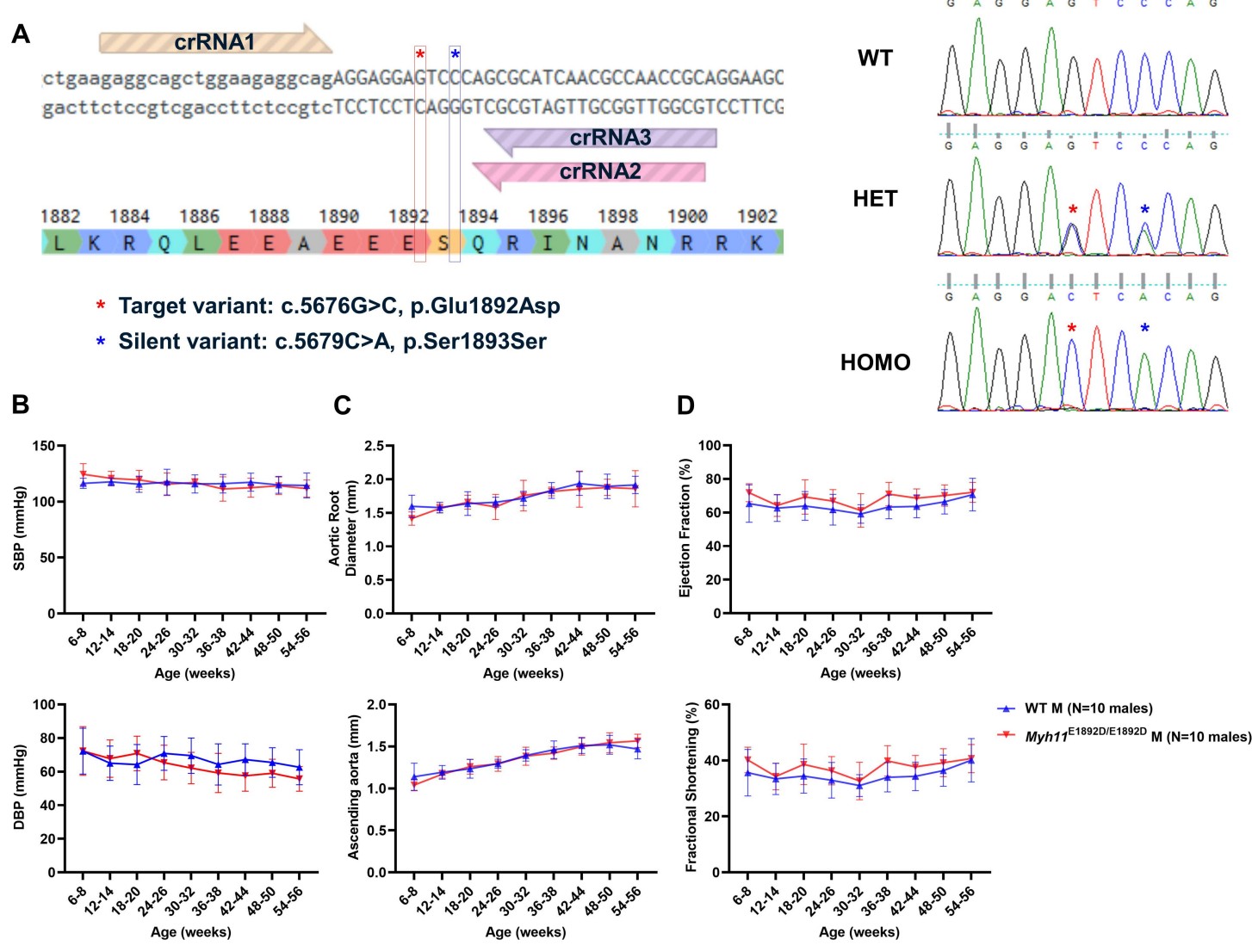

**Fig 1. Cardiovascular phenotyping in male mice. (A)** Generation of the $Myh11^{E1892D/E1892D}$ mouse model using CRISPR-Cas9 genome editing. Sequencing of DNA sample from tail tissue and cDNA sample from both heart and aorta confirms single nucleotide variant in mouse $Myh11$ gene. **(B)** Tail-cuff blood pressure measurement. **(C)** and **(D)** Echocardiographic measurements of aortic root, ascending aorta, and left ventricular contractile function. N = 10 males in each group. SBP, systolic blood pressure; DBP, diastolic blood pressure; WT, wild-type; HET, heterozygous; HOMO, homozygous.

aorta did not differ between the $Myh11^{E1892D/E1892D}$ and WT mice, with the exception of significant enlargement of ascending aorta in older female $Myh11^{E1892D/E1892D}$ mice compared to female WT mice (Figs 1B, 1C, S1A and S1B). Left ventricular contractile function was assessed, and similar ejection fraction and fractional shortening were observed in the WT and mutant mice (Figs 1D, S1C).

To evaluate SMC contractility in the aortas, the isometric force of ascending aortic rings in response to contractile agonists and vasodilators was measured using aortas from male WT and $Myh11^{E1892D/E1892D}$ mice at 10 months of age. The contractile tension development and the maximum force generation in response to phenylephrine or potassium chloride showed no difference between $Myh11^{E1892D/E1892D}$ and WT aortas (**Fig 2A**). A similar level of arterial relaxation was also

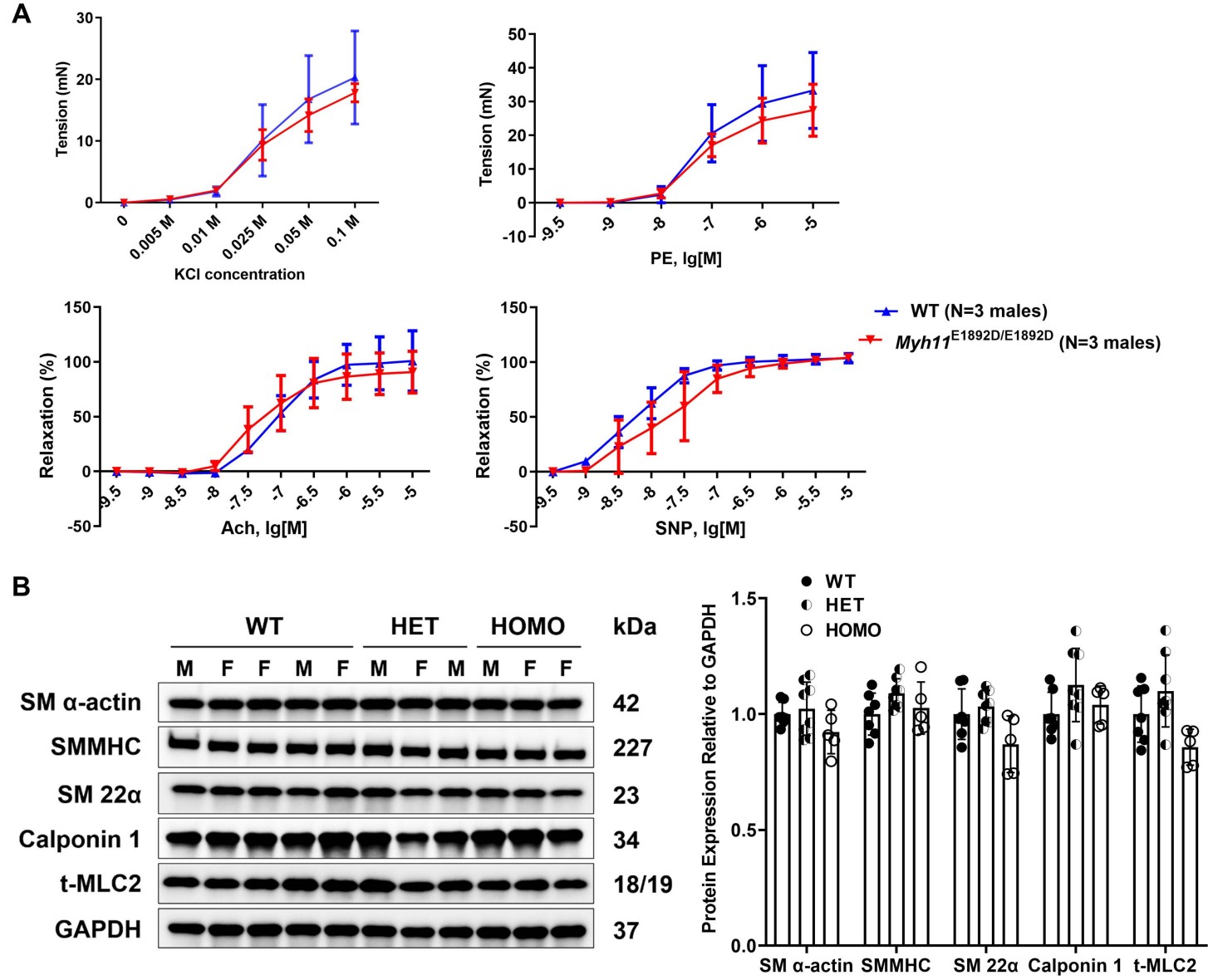

**Fig 2. Assessment of myograph and smooth muscle cell contractile protein expression in ascending aortic tissues. (A)** Myographic assay of mouse ascending aortic rings at 10 months of age. N = 3 males in each group. **(B)** Immunoblot assay of protein lysates of the ascending aortas from wild-type (WT), heterozygous (HET), and homozygous (HOMO) mice at the age of 6 months. N = 7 (3M + 4F), 8 (5M + 3F), 5 (2M + 3F) in the WT, HET, and HOMO groups, respectively. KCl, potassium chloride; PE, phenylephrine; Ach, acetylcholine; SNP, sodium nitroprusside.

found in response to acetylcholine or sodium nitroprusside (**Fig 2A**). Finally, immunoblot analyses of protein lysates of the ascending aortas showed no difference in SMC contractile markers among the WT, *Myh11*E1892D/+, and *Myh11*E1892D/E1892D aortas (**Fig 2B**).

## Pressure overload augments ascending aortic enlargement in *Myh11*E1892D/E1892D mice

We previously demonstrated that the proximal aorta enlarges two weeks after TAC in WT C57BL/6J mice and is associated with aortic medial and adventitial thickening [24]. *Myh11*E1892D/E1892D and WT mice of both sexes were subjected to TAC

surgeries at 10–12 weeks of age. Mortality rates immediately following recovery from anesthesia were similar across the four groups: 22% (2/9) for male WT, 30% (3/10) for male mutants, 30% (3/10) for female WT, and 20% (2/10) for female mutants. These deaths were associated with acute congestive heart failure due to the constriction. Additionally, one female mutant mouse died of ruptured left main coronary artery and cardiac tamponade one day after TAC, and one male mutant mouse died of congestive heart failure on day eight (S2 Fig and S2 Table). Two weeks post-surgery, both male and female *Myh11*E1892D/E1892D mice exhibited significant increases in the ascending aortic diameter compared to sex-matched WT TAC mice, despite displaying comparable levels of systolic and diastolic blood pressure (Figs 3A, 3B, S3A and S3B). Histological analysis revealed significant increases in medial thickening and the number of elastic breaks, and decreased medial cell density in the mutant aortas of both sexes compared to WT aortas, with no difference in adventitial collagen accumulation (Figs 3A, 3C, S4A and S4B).

### Pressure overload induces left ventricular posterior wall hypertrophy and heart failure in male *Myh11*E1892D/E1892D mice

TAC increases cardiac afterload and is routinely used to study cardiac hypertrophy and heart failure [25]. Unexpectedly, male *Myh11*E1892D/E1892D mice undergoing TAC had significantly impaired left ventricular contractile function by echocardiography two weeks after TAC, as illustrated by decreased ejection fraction, stroke volume, fractional shortening, and cardiac output in these mutant mice compared to similarly treated male WT mice (Fig 4A and 4B); these changes were not present post TAC in the female mutant mice compared to female WT mice (S3C Fig). Subsequent evaluation revealed that TAC induced a significant increase in left ventricular mass in male *Myh11*E1892D/E1892D mice compared to male WT TAC mice, primarily characterized by posterior wall thickening (Fig 4C). Additionally, both end-systolic and end-diastolic diameters and volumes of the left ventricle were significantly enlarged in male *Myh11*E1892D/E1892D mice (Fig 4C). In contrast, alterations of these cardiac remodeling were not observed in female *Myh11*E1892D/E1892D mice after TAC, except those limited exclusively to the end-diastolic thickness of the left ventricular anterior wall when compared to WT female mice (S3D Fig).

Heart tissue obtained from male WT and *Myh11*E1892D/E1892D mice post-TAC was sectioned and stained to assess cardiomyocyte hypertrophy and fibrosis [26,27]. The cardiomyocyte cross-sectional area in the posterior wall of the left ventricle was increased significantly in the *Myh11*E1892D/E1892D heart compared to WT hearts, thus confirming cardiomyocyte hypertrophy, while no difference in cardiomyocyte diameter in the anterior walls was observed (Figs 4D and S5). Quantification of collagen deposition showed increased peri-arterial and left ventricular posterior wall fibrosis in the *Myh11*E1892D/E1892D hearts compared to WT hearts (Fig 4E), which was not observed in female mice (S4A and S4B Fig).

### Discussion

A heterozygous missense VUS in the coiled-coil domain of *MYH11*, p.Glu1892Asp, was identified in a proband with aortic root aneurysm. The functional impact of this variant was investigated by introducing this variant into the mouse genome and generating homozygous rather than heterozygous mutant mice due to the fact that almost all heterozygous pathogenic variants in genes predisposing to TAD identified in humans require homozygosity in mice to cause disease [14,28]. Similar to other mouse models of *MYH11* genetic variants [21,23], *Myh11*E1892D/E1892D mice develop normally without thoracic aortic enlargement, but increasing the forces on the aorta via TAC augments ascending aortic enlargement in *Myh11*E1892D/E1892D mice compared to similarly treated WT mice. *MYH11*, p.K1256del, is a pathogenic variant that causes an autosomal dominant inheritance of a predisposition for type A and B dissections [14]. Although there is no evidence of TAD in mice heterozygous or homozygous for this variant, angiotensin II infusion induces both thoracic and abdominal aortic dissections in both heterozygous and homozygous mice [23]. Thus, these data support that the *MYH11*, p.Glu1892Asp, variant increases the risk for TAD, but further data are needed to determine the penetrance and additional genetic or environmental factors that contribute to the penetrance of TAD associated with this variant.

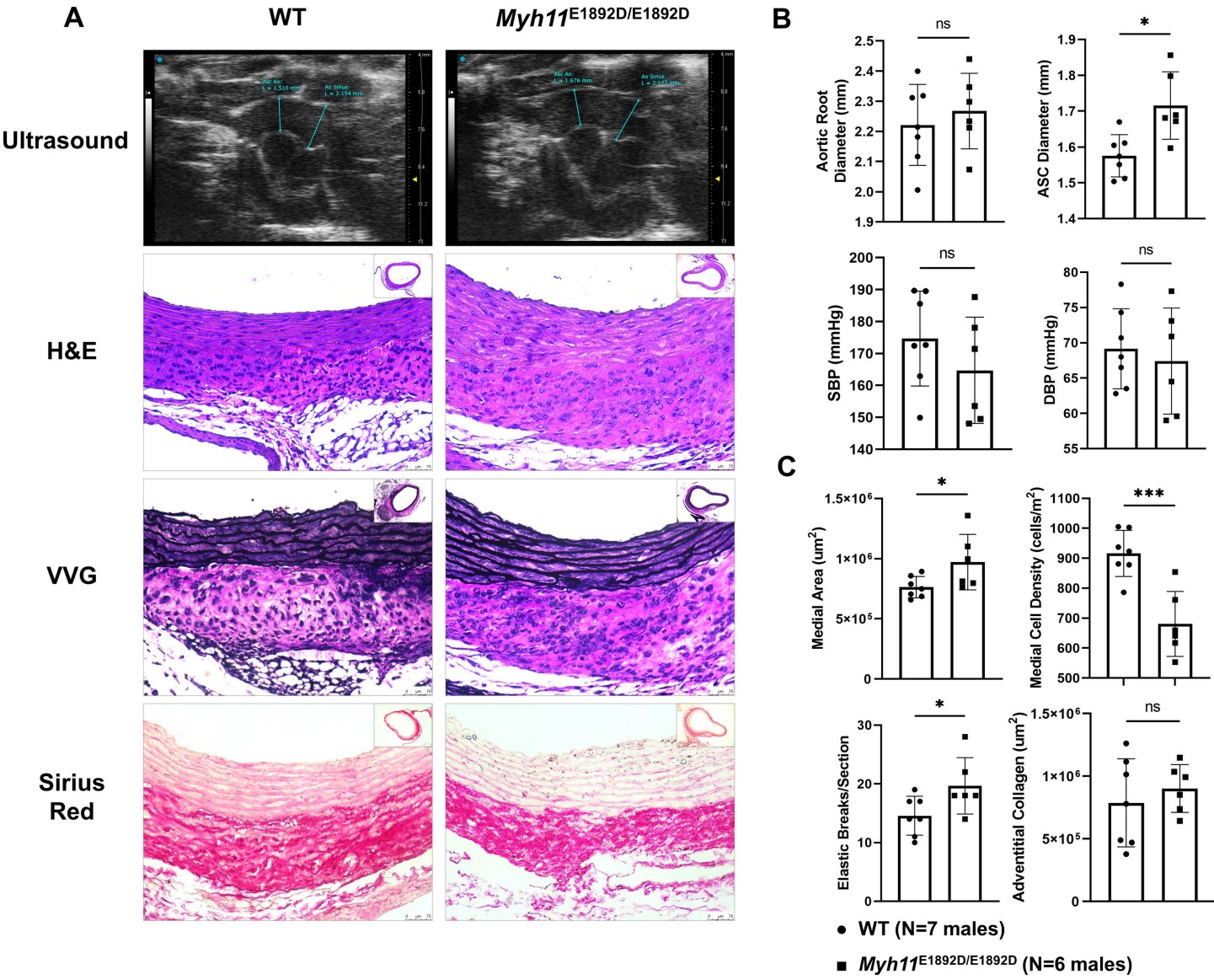

**Fig 3. Assessment of ascending aortic remodeling 2 weeks after transverse aortic constriction (TAC) in male mice. (A)** Representative images of proximal aortic ultrasound measurement, H&E, VVG, and Sirius Red staining on ascending aortic tissue sections. **(B)** TAC induces similar levels of systolic (SBP) and diastolic (DBP) blood pressure in wild-type (WT) and *Myh11*E1892D/E1892D mice, along with significant ascending aortic enlargement in *Myh11*E1892D/E1892D mice. **(C)** Histology analysis shows significantly increased medial thickening and elastic breaks, and decreased medial cell density in the mutant aortas compare with WT TAC aortas, with no difference of adventitial collagen accumulation. N = 7 and 6 in the WT and *Myh11*E1892D/E1892D groups, respectively. ns, non-significant; * $P < 0.05$, *** $P < 0.001$, by unpaired Mann-Whitney analysis.

An unexpected finding in this study is the sex-dependent aberrant cardiac remodeling observed in *Myh11*E1892D/E1892D mice with pressure overload, as evidenced by the increased cardiomyocyte hypertrophy, posterior wall thickening, and left ventricle failure following TAC in male but not female mice. TAC is a well-established model to mimic hypertensive heart failure in humans, particularly replicating cardiac hypertrophy and subsequent heart failure [29]. *MYH11* expression is the most specific marker of SMCs identified to date and it is not expressed in other cell types, including myofibroblasts [5–9]. Thus, our data indicate that a rare variant in a gene expressed exclusively in SMCs can trigger increased cardiomyocyte

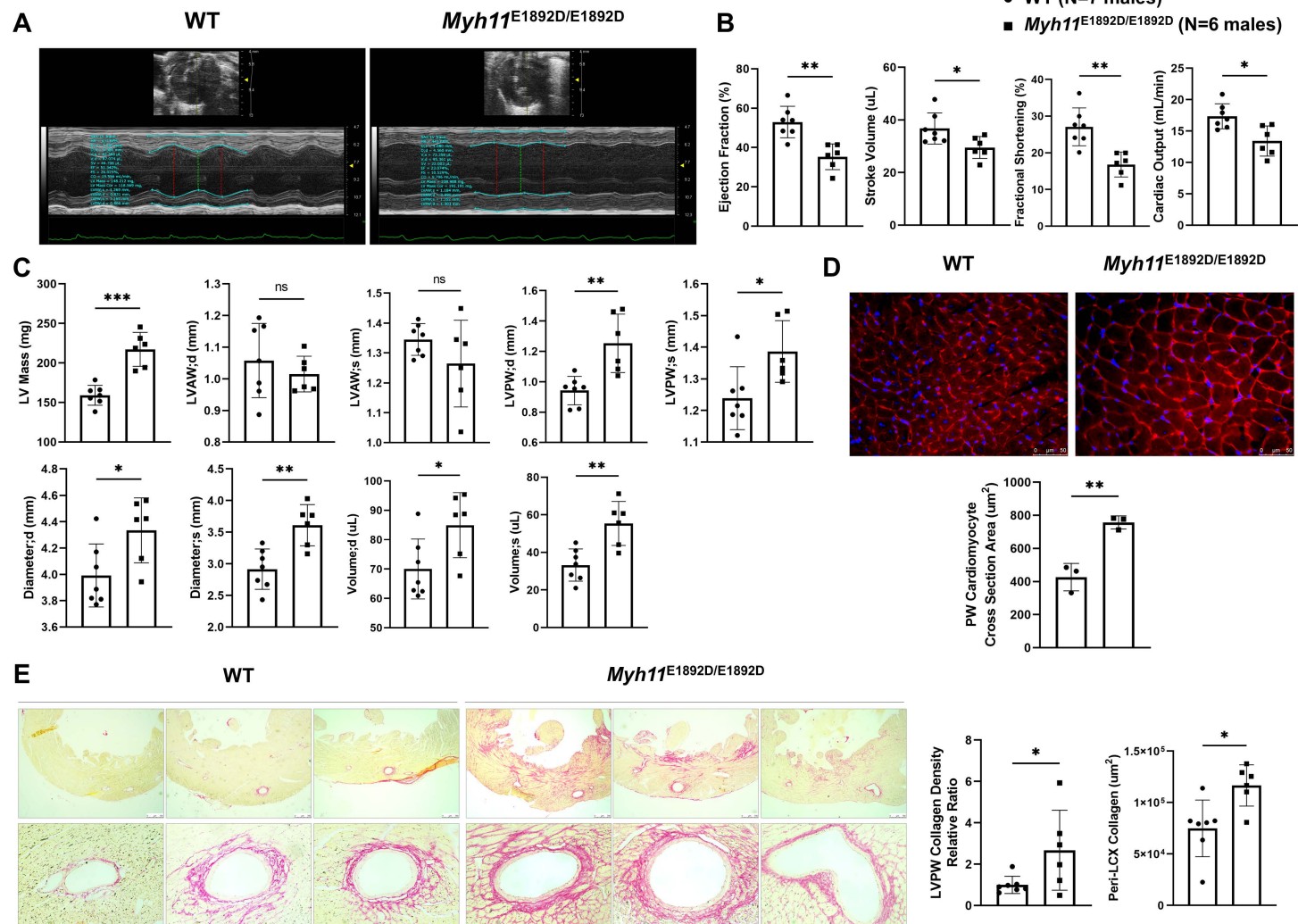

**Fig 4. Assessment of left ventricular (LV) remodeling 2 weeks after transverse aortic constriction (TAC) in male mice. (A)** Representative images of LV contraction measurement from short-axis parasternal view using M-mode. **(B)** Four functional parameters show decreased LV contractility in *Myh11*[E1892D/E1892D] mice 2 weeks after TAC. **(C)** Structural parameters of LV show increased end diastole (d) and systole (s) thickness of posterior wall (PW), along with increased LV diameters and volumes in *Myh11*[E1892D/E1892D] mice after TAC. **(D)** Wheat Germ Agglutinin (WGA) staining of LVPW shows significant increase of cardiomyocyte cross section area in male *Myh11*[E1892D/E1892D] heart after TAC. **(E)** Representative images of Sirius Red staining of LVPW. Quantification of collagen deposition area shows increased peri-arterial (left circumflex artery, LCX) area and LVPW collagen density in *Myh11*[E1892D/E1892D] heart after TAC. N = 7 and 6 in the WT and *Myh11*[E1892D/E1892D] groups, respectively. ns, non-significant; * *P* < 0.05, ** *P* < 0.01, *** *P* < 0.001, by unpaired Mann-Whitney analysis.

hypertrophy and decreased left ventricular contractile function, implicating a novel role for arterial SMCs in driving pathologic cardiac remodeling with pressure overload. This finding contrasts with pathogenic variants in *FBN1*, which encodes fibrillin-1, a major component of extracellular matrix microfibrils. *FBN1* mutations cause Marfan syndrome, a genetic disorder characterized by TAD, skeletal, and ocular abnormalities. While pathogenic variants in *FBN1* are also associated with an increased risk for dilated cardiomyopathy in both patients and mice, *FBN1* is expressed in many cell types, including SMCs, cardiomyocytes, and cardiac fibroblasts [30], making it challenging to determine which cell type is primarily responsible for cardiomyopathy.

In this model, constriction of the transverse aorta leads to increased pressure load on the left ventricle, triggering a cascade of molecular events similar to those observed in clinical conditions such as poorly controlled hypertension or aortic stenosis. Studies utilizing the TAC mouse model to study the mechanisms underlying cardiac remodeling and pump failure need to consider the genetic background [31,32], degree and duration of constriction [33,34], and sex [35,36]. Male C57BL/6J mice undergoing TAC using a 27-gauge needle develop cardiac hypertrophy and pump failure as early as 7 days after surgery, which are characterized by increased mass of left ventricle, thicknesses of septal and posterior wall, along with decreased ejection fraction and fractional shortening [33]. In the current study, we replicate the decreased left ventricular contractile function in male WT mice 2 weeks after surgery using a 27-gauge needle [24], and identify further decline of heart contraction in male $Myh11^{E1892D/E1892D}$ mice. It has been reported that TAC-induced cardiac hypertrophy and impaired contraction show sex differences 6 weeks after TAC in C57BL/6J WT mice [36]. However, in this study, both ejection fraction and fractional shortening are significantly decreased in male versus female $Myh11^{E1892D/E1892D}$ mice just 2 weeks after surgery, indicating a rapid decline of left ventricular contractile function in male $Myh11^{E1892D/E1892D}$ mice (S6 Fig). When female mice lacking estrogen receptor beta gene ($Esr2^{-/-}$) are subjected to TAC for 2 weeks, a greater increase in heart weight relative to body weight is observed compared to WT littermate females [35]. This finding suggests that estrogen receptor subtype beta plays a protective role in the development of pressure overload-induced cardiac hypertrophy and may be the mediator of observed sex differences.

We hypothesize that $Myh11^{E1892D/E1892D}$ SMCs in coronary arteries may produce a signal that alters the cardiomyocyte. We previously identified increased IGF-1 expression in aortic tissue of a patient with $MYH11$ p.Leu1264Pro, another missense variant in the coiled-coil domain [13]. Transcriptomic analyses identified a 40-fold increase of $IGF1$ expression in the SMCs explanted from the patient's aorta compared to SMCs explanted from normal aortas. Thus, $MYH11$ rare variants may trigger excessive SMC IGF-1 production and be the source of SMC-to-cardiomyocyte signaling to drive cardiac hypertrophy and failure in $Myh11^{E1892D/E1892D}$ mice after TAC. One of the main pathways activated by IGF-1 is the PI3K/Akt/mTORC1 pathway that promotes protein synthesis and cell growth, contributing to cardiomyocyte hypertrophy in response to pressure overload. Inhibition of mTORC1 with rapamycin significantly attenuates cardiac hypertrophy and improves cardiac function with pressure overload [37,38].

Another interesting finding in this study is that pressure overload leads to increased cardiac fibrosis in the posterior wall of the $Myh11^{E1892D/E1892D}$ hearts, characterized by increased peri-arterial and interstitial collagen deposition. Pressure overload-induced cardiac fibrosis is an intricate process influenced by various molecular mechanisms, with activated fibroblasts and myofibroblasts acting as the central effectors and serving as the main source of matrix proteins. One key driver is the activation of the renin-angiotensin-aldosterone system due to decreased stroke volume and renal blood flow [39], facilitating fibroblast proliferation and collagen deposition in the myocardium [40,41]. Increased biomechanical stress on the cardiac tissue leads to release and activation of transforming growth factor-beta signaling, stimulating fibroblast differentiation into myofibroblasts, which are responsible for excessive extracellular matrix production [42,43]. The activation of profibrotic pathways, such as the renin-angiotensin-aldosterone system and transforming growth factor-beta signaling, is evident in TAC-induced cardiac remodeling [41,44]. Inflammatory responses mediated by cytokines and immune cells also contribute to the progression of fibrosis post-TAC [41], while oxidative stress and mitochondrial dysfunction have been implicated in TAC-induced cardiac fibrosis and dysfunction [45,46]. Additionally, fibroblasts can also become activated by mechanical stress through mechanosensitive receptors like integrins, ion channels, G-protein coupled receptors, and growth factor receptors and can activate downstream signaling pathways that promote matrix production [47]. Further studies will define the predominant signaling pathway that mediates the rapid interstitial collagen deposition in male $Myh11^{E1892D/E1892D}$ mice.

Genome-wide association studies (GWAS) have identified loci involving $MYH11$ associated with various traits of cardiac rhythm, including resting heart rate, heart rate response to exercise, atrial fibrillation, PR interval, and electrocardiography, suggesting a potential relationship between $MYH11$ variants and cardiac pacing and arrhythmias. Additionally, three

genetic risk loci (rs216158, rs9972711, rs12691049) encompassing *MYH11* have been linked to coronary artery disease [48]. Notably, no loci linked to *MYH11* have been associated with cardiac hypertrophy or heart failure in GWAS.

Collectively, these results demonstrate that a missense VUS in a gene almost exclusively expressed in SMCs, *MYH11*, does indeed increase thoracic aortic enlargement but also triggers aberrant pressure overload-induced remodeling of the heart that is characterized by increased cardiomyocyte hypertrophy, cardiac fibrosis, and heart failure in males. These findings provide further evidence of the diverse phenotypes associated with *MYH11* rare variants and implicate vascular SMC-to-cardiomyocyte signaling in driving aberrant cardiac remodeling with pressure overload. Future studies will focus on the interactions among different cell types in the heart and identify specific cellular pathways downstream of the mutant contractile protein in SMCs that mediate SMC-cardiomyocyte communications and contribute to cardiomyopathy.

## Materials and methods

### Ethics statement

All animal experimental procedures were designed in accordance with National Institutes of Health guidelines and approved by the Animal Welfare Committee and the Center for Laboratory Animal Medicine and Care at the University of Texas Health Science Center at Houston (AWC-24–0086).

### Animal study

*Myh11*[E1892D/+] breeders were transferred from the Jackson Laboratory and the colony was maintained on a C57BL/6J background. The mutant *Myh11* mouse allele was generated via direct delivery of CRISPR-Cas9 reagents into mouse zygotes. Site-directed mutagenesis was used to introduce a nucleotide substitution (c.5676G>C) in the *Myh11* gene (ENSMUSG00000018830), corresponding to the Ensembl transcript ENSMUST00000230397.1. This modification resulted in an alteration of codon 1892 from glutamic acid (E) to aspartic acid (D) (p.E1892D). To prevent re-cutting by CRISPR reagents, a synonymous c.5679C>A variant was engineered as a silent mutation. Off-target effects were assessed using a stringent prediction criterion and no off-target editing was detected (S3 Table). DNA sequencing confirmed the target editing and the presence of c.5679C>A variant in the founder and their co-segregation in offspring after crossing with wild-type C57BL/6J mice (Fig 1A).

### Transverse aortic constriction surgery

At the age of 10–12 weeks, both male and female wild-type and *Myh11*[E1892D/E1892D] mice were anesthetized by 0.3-0.5 L/min pure oxygen with 2% isoflurane and placed supine on a 38°C heating pad. Intubation was performed with a 22-gauge venous catheter connected to a rodent ventilator with a respiratory rate of 125–150 breaths/min and a tidal volume of 6–8 µL/g. Carprofen (dose of 5 mg/kg, subcutaneous injection) and lidocaine (dose<2.25 mg/kg, subcutaneous injection) were administered before an upper partial sternotomy incision (about 1 cm) was made. A 6–0 silk suture was coiled under the aortic arch between the innominate artery and the left common carotid artery and ligated with a 27-gauge needle placed by the aortic arch. The needle was then promptly removed to yield a constriction of 0.41mm in the outer diameter. The lungs were re-inflated before the skin was closed. Mice that died prior to the endpoint at fourteen days post-operation were subjected to necropsy to determine the cause of death. In male and female mice, comparable numbers succumbed following surgery (males: 2 out of 9 WT and 3 out of 10 mutants; females: 3 out of 10 WT and 2 out of 10 mutants). These expected post-operative mortality rates are primarily linked to acute congestive heart failure post-TAC [25], with the exception of one female that died due to coronary artery rupture (S2 Fig and S2 Table).

### Echocardiography

Echocardiography (Vevo 3100 imaging system, MX550D transducer, VisualSonics, Toronto, Canada) was performed two weeks post-surgery. Briefly, mice were weighed and anesthetized by 0.5-1.0 L/min room air with 2% isoflurane via nose

cone. Heart rate was closely monitored and body temperature was maintained around 38.5°C using the heating system. The aortic root and ascending aorta were imaged in B-mode. Left ventricular function derived from short axis parasternal planes was imaged in M-mode. Measurements of the maximal internal diameter of the proximal aorta and left ventricular contractile function were obtained from three different cardiac cycles and averaged. Data were analyzed by an operator blinded to the treatment groups.

## Invasive blood pressure measurement

Following echocardiography analyses, intraluminal blood pressure measurements were performed using a Millar pressure catheter (SPR-1000, 1.0F, Oakville, Ontario, Canada) inserted into the right common carotid artery. Mice were intubated and placed on a ventilator using the same conditions as in TAC surgery except replacing pure oxygen with room air. The 1.0F catheter was inserted into the ascending aorta to monitor the blood pressure. Stable pressure tracings were recorded for 5 minutes at a PCU-2000 pressure signal conditioner and PowerLab 4/35 station (ADInstruments Inc., Colorado Springs, CO, USA), and systolic and diastolic blood pressures were averaged from a stable 4-minute recording period.

## Myographic assay of aortic rings

Ascending aortic tissues were harvested from both male and female mice at the age of 10 months and delivered in ice-cold Hanks' Balanced Salt Solution through overnight shipping, and then cut into 2-mm ring segments and placed in the 620M Multi Chamber Myograph System (Danish Myo Technology, Hinnerup, Denmark) filled with 8 mL of oxygenated (95% $O_2$, 5% $CO_2$) physiological saline solution (118.31 mM NaCl, 4.69 mM KCl, 1.2 mM $MgSO_4$, 1.18 mM $KH_2PO_4$, 24.04 mM $NaHCO_3$, 0.02 mM EDTA, 2.5 mM $CaCl_2$, and 5.5 mM glucose) and allowed to equilibrate at 37 °C for at least 30 min. Aortic rings were stretched in 2–4 mN increments from 0 mN until the calculated transmural pressure reached 13.3 kPa (100 mmHg). Optimal resting tension was applied to the rings based on the passive vascular length-tension relationship. Cumulative concentration-response curves to phenylephrine (PE, $10^{-9}$ to $10^{-5}$ M) and potassium chloride (KCl, 5–100 mM) were generated to assess contractile function. Vascular relaxation was assessed with acetylcholine (Ach, $10^{-9}$ to $10^{-5}$ M) and sodium nitroprusside (SNP, $10^{-9}$ to $10^{-5}$ M) administration. Concentration-response curves to Ach and sodium nitroprusside were generated after rings were pre-constricted to 70% of maximum with PE. Doses were added after the response curve reached a plateau from the previous dose. Percent vasocontractile responses (%) were calculated for PE and KCl as $[(D_P - D_B)/D_B] \times 100$, where '$D_P$' is the maximal force generated by a given specific dose and '$D_B$' is the baseline force. Percent relaxation responses were calculated as $[(D_P - D_D)/(D_P - D_B)] \times 100$, where $D_P$ is the maximal force pre-generated by PE, $D_D$ is the lowest force generated at a given dose of ACh or SNP and $D_B$ is the baseline force [49,50].

## Histopathology

After intraperitoneal injection with Avertin (2.5%, 350 mg/kg), euthanized animals were perfusion fixed with 20 mL 1 × PBS (pH = 7.4) followed by 20 mL 10% neutral buffered formalin for 5 minutes through the left ventricle under physiological pressure. Ascending aortas and heart tissues were excised and further fixed in 10% neutral buffered formalin overnight at room temperature, then embedded in paraffin and sectioned at 5 μm. Aortic sections were stained with hematoxylin and eosin (H&E), Verhoeff Van Gieson (VVG, Polysciences, Inc., 25089–1), and Picro-Sirius Red (Abcam, ab150681) for morphometric analyses, medial elastic fibers, and collagen content identification, respectively. Heart sections were stained with Picro-Sirius Red to determine collagen content. Images were obtained using a Leica DM2000 LED microscope, and analyzed with ImageJ software. Elastic fiber breaks are defined as fragmentation or loss of continuity in the black-stained elastin fibers. For collagen deposition analysis, high-resolution TIF images were processed by splitting them into their red, green, and blue grayscale components. The green channel was selected for further analysis, as it provided the best contrast between Sirius Red-stained collagen and the background. A thresholding technique was then applied to the green

channel to quantify collagen-rich areas. At least 3 representative cross sections per aortic sample were analyzed and averaged. Quantitative analyses were performed by three individuals blinded to the group information.

## Wheat germ agglutinin staining

After rehydration, heart sections were stained with CF@640 dye WGA solution (Biotium, #29026–1) for 20 minutes at room temperature and protected from light, then mounted with DAPI (VECTASHIELD Antifade Mounting Medium, H-1200). Immunofluorescent images were obtained using the Leica DMi8 confocal microscope and analyzed with ImageJ software.

## Immunoblot analyses

Proximal aortic tissue lysates were collected from ≥ 5 biological replicates per condition. Lysates were fractionated by SDS-PAGE and transferred to a polyvinylidene difluoride membrane according to standard protocols. Immunoblot images were quantitated with ImageJ software.

## Statistical analysis

Data are presented as mean ± standard deviation. Nonparametric statistical tests were conducted. Statistical differences between two groups were analyzed using unpaired Mann-Whitney U test. Analyses were performed using GraphPad Prism 9.0. Statistical significance was set at $P$-value < 0.05.

## Supporting information

**S1 Fig. Cardiovascular phenotyping in female mice. (A)** Tail-cuff blood pressure measurement. **(B)** and **(C)** Echocardiographic measurements of aortic root, ascending aorta, and left ventricular contractile function. N = 10 females in each group. SBP, systolic blood pressure; DBP, diastolic blood pressure; WT, wild-type. * $P$ < 0.05, by unpaired Mann-Whitney analysis.
(TIF)

**S2 Fig. Necropsy of a mouse died one day after transverse aortic constriction.** One $Myh11^{E1892D/E1892D}$ female mouse died of ruptured left main coronary artery and associated cardiac tamponade one day after TAC. Yellow arrow and a 5–0 suture show the rupture site.
(TIF)

**S3 Fig. Echocardiography and central blood pressure measurements 2 weeks after transverse aortic constriction (TAC) in female mice. (A)** Aortic root and ascending (ASC) aortic diameters. **(B)** Systolic (SBP) and diastolic (DBP) blood pressures. **(C)** Evaluation of left ventricular (LV) contractile function in female mice after TAC. **(D)** Structural evaluation of LV in female mice after TAC. N = 7 females in each group. AW, anterior wall; PW, posterior wall; d, end diastolic; s, end systolic; ns, non-significant; * $P$ < 0.05, by unpaired Mann-Whitney analysis.
(TIF)

**S4 Fig. Histopathological analysis of ascending aorta and heart tissue 2 weeks after transverse aortic constriction (TAC) in female mice. (A)** Representative images of H&E, VVG, and Sirius Red staining on ascending aortic tissue and left ventricular posterior wall. **(B)** Histology analysis shows significantly increased medial elastic breaks and decreased medial cell density in the mutant aortas compare to WT TAC aortas, with no difference of medial area or collagen accumulation in the aortic adventitia and left ventricular posterior wall. N = 7 females in each group. ns, non-significant; * $P$ < 0.05, ** $P$ < 0.01, by unpaired Mann-Whitney analysis.
(TIF)

**S5 Fig. Wheat Germ Agglutinin (WGA) staining of left ventricular anterior wall (LVAW) in male mice 2 weeks after transverse aortic constriction (TAC).** There is no difference of cardiomyocyte cross-sectional area between wild-type and *Myh11*^E1892D/E1892D^ mice after TAC. N = 3 in each group. ns, non-significant, by unpaired Mann-Whitney analysis.
(TIF)

**S6 Fig. Comparison of left ventricular contractile function between male and female mutant mice 2 weeks after transverse aortic constriction (TAC).** Male *Myh11*^E1892D/E1892D^ mice exhibit significantly lower ejection fraction and fractional shortening compared to female mutant mice following TAC. N = 6 and 7 in the male and female groups, respectively. ns, non-significant; ** $P<0.01$, by unpaired Mann-Whitney analysis.
(TIF)

**S1 Table. Segregation record of 102 progenies from heterozygous breeders.**
(XLSX)

**S2 Table. Total numbers of mice died after transverse aortic constriction.**
(XLSX)

**S3 Table. Off-target analysis of CRISPR-Cas9 genome editing.** Potential off-target effects were assessed using a stringent prediction criterion: off-target sites with a score <1.5 for canonical protospacer adjacent motif (5'-NGG-3') or <2.0 for non-canonical protospacer adjacent motif (5'-NAG-3') were analyzed. No low-frequency off-target editing was detected. crRNA, CRISPR RNA.
(XLSX)

## Acknowledgments

We would like to thank the family for participating in this study. We also gratefully acknowledge the contributions of Nicholas Brown, Hongwei Jin, and the Center for Biometric Analysis at The Jackson Laboratory for expert assistance with this project.

## Author contributions

**Conceptualization:** Zhen Zhou, Jun Wang, John M. Greally, Dianna M Milewicz.

**Data curation:** Zhen Zhou, Kgosi Hughes, Nisha Saif, Hyoseon Kim, Michael P. Massett, Mingjie Zheng, Alana C. Cecchi, Dongchuan Guo, David R. Murdock, Ping Pan, Jelita S. Clinton.

**Formal analysis:** Zhen Zhou, Kgosi Hughes, Nisha Saif, Hyoseon Kim, Michael P. Massett, Mingjie Zheng, Alana C. Cecchi, Dongchuan Guo, David R. Murdock.

**Funding acquisition:** Dianna M Milewicz.

**Investigation:** Zhen Zhou, Kgosi Hughes, Nisha Saif, Hyoseon Kim, Michael P. Massett, Mingjie Zheng, Alana C. Cecchi, Dongchuan Guo, David R. Murdock.

**Methodology:** Zhen Zhou, Kgosi Hughes, Nisha Saif, Hyoseon Kim, Michael P. Massett, Mingjie Zheng, Alana C. Cecchi, Dongchuan Guo, David R. Murdock, Ping Pan, Jelita S. Clinton.

**Project administration:** Zhen Zhou.

**Resources:** Michael P. Massett, Jun Wang, John M. Greally, Dianna M Milewicz.

**Supervision:** Dianna M Milewicz.

**Writing – original draft:** Zhen Zhou.

**Writing – review & editing:** Zhen Zhou, Kgosi Hughes, Nisha Saif, Hyoseon Kim, Michael P. Massett, Mingjie Zheng, Alana C. Cecchi, Dongchuan Guo, David R. Murdock, Ping Pan, Jelita S. Clinton, Jun Wang, John M. Greally, Dianna M Milewicz.

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
