## [Decision Letter · Decision Letter 0]

Dear Dr Milewicz,

Thank you very much for submitting your Research Article entitled 'MYH11 rare variant augments aortic growth and induces cardiac hypertrophy and heart failure with pressure overload' to PLOS Genetics.

The manuscript was fully evaluated at the editorial level and by independent peer reviewers. The reviewers appreciated the attention to an important problem, but raised some substantial concerns about the current manuscript. Based on the reviews, we will not be able to accept this version of the manuscript, but we would be willing to review a much-revised version. We cannot, of course, promise publication at that time.

If you decide to revise the manuscript for further consideration at PLOS Genetics, please aim to resubmit within the next 60 days, unless it will take extra time to address the concerns of the reviewers, in which case we would appreciate an expected resubmission date by email to plosgenetics@plos.org.

If present, accompanying reviewer attachments are included with this email; please notify the journal office if any appear to be missing. They will also be available for download from the link below. You can use this link to log into the system when you are ready to submit a revised version, having first consulted our Submission Checklist .

PLOS has incorporated Similarity Check , powered by iThenticate, into its journal-wide submission system in order to screen submitted content for originality before publication. Each PLOS journal undertakes screening on a proportion of submitted articles. You will be contacted if needed following the screening process.

To resubmit, log into your Editorial Manager account and select the option 'Revise Submission' in the 'Submissions Needing Revision' folder.

We are sorry that we cannot be more positive about your manuscript at this stage. Please do not hesitate to contact us if you have any concerns or questions.

Yours sincerely,

Chad S. Weldy, M.D., Ph.D.

Guest Editor

PLOS Genetics

Hua Tang

Section Editor

PLOS Genetics

Dear Dr. Milewicz,

Thank you for your submission. I have had 4 experts in this field evaluate your submitted manuscript. As you can see from the comments below, there is consensus that there is merit to this work and may be a good fit for Plos Genetics. However, there is agreement between several of the reviewers of a need to discuss the discrepancy between the frequency of the variant in the general population (relatively common) and clinical phenotype, providing more detail on the proband (ie. heterozygous/homozygous) and relationship to the murine phenotype, and better discussion, and likely need for more molecular experiments, to understand the mechanism of effect on myocardial function. My recommendation is for resubmission with major revisions.

Best,

Chad Weldy, MD, PhD

Guest Editor

Stanford University

Reviewer's Responses to Questions

**Comments to the Authors:**

Reviewer #1: Zhou et al investigated the effects of the MYH11 p.Glu1892Asp variant, a rare variant of uncertain significance, on cardiovascular phenotypes in mice. The authors used genomic editing to create mice with this specific variant (Myh11E1892D/E1892D) and subjected them to transverse aortic constriction (TAC) to simulate pressure overload. While these mutant mice exhibited normal cardiovascular parameters under baseline conditions, TAC induced significant aortic enlargement, elastic fragmentation, and cardiac hypertrophy, particularly in male mice. These male mutant mice also showed impaired left ventricular function, reduced ejection fraction, stroke volume, and increased left ventricular mass due to posterior wall thickening. The findings suggest that the MYH11 variant contributes to adverse cardiovascular remodeling and heart failure under pressure overload conditions, highlighting the importance of smooth muscle cells in these processes.

This manuscript presents a well-structured and thorough investigation into the effects of a rare MYH11 variant on cardiovascular remodeling under pressure overload conditions. The study is commendable for its use of advanced genomic editing techniques to create a relevant mouse model, providing valuable insights into the functional consequences of the MYH11 p.Glu1892Asp variant. The experimental design is robust, with comprehensive cardiovascular phenotyping and detailed histological analyses demonstrating the variant's impact on aortic enlargement and cardiac hypertrophy. The authors also highlight important sex-specific differences in the response to pressure overload, adding a novel dimension to the understanding of MYH11-related pathologies.

The only thing missing from the manuscript is the authors need to indicate if the rare variant found in the proband is homozygous or heterozygous. Otherwise, this reviewer believes that the manuscript should be accepted with this minor modification.

Reviewer #2: In this manuscript, the authors investigate the impact of a rare variant (p.Glu1892Asp) in the MYH11 gene identified in a patient with an aortic root dilatation using a knock-in mouse model. Through genomic editing, the authors generated Myh11E1892D/E1892D mice and performed cardiovascular assessments on mutant and wild-type (WT) controls. The results indicate that mutant mice of both sexes were largely comparable to WT mice in terms of growth, blood pressure, and heart function up to 13 months of age, with the exception of significant enlargement of the ascending aorta in older female Myh11E1892D/E1892D mice compared to female WT controls. The mice were then subjected to transverse aortic constriction (TAC) to induce hypertension. While TAC increased blood pressure in both groups, mutant mice showed a greater degree of ascending aortic enlargement and elastic fiber fragmentation. In male Myh11E1892D/E1892D mice, there was a notable reduction in ejection fraction, stroke volume, fractional shortening, and cardiac output compared to male WT mice two weeks after TAC. Additionally, left ventricular mass significantly increased, primarily due to posterior wall thickening, with cardiomyocyte hypertrophy and elevated collagen deposition in the myocardium and surrounding arteries. The authors conclude that these findings highlight the sex-specific clinical variability associated with the rare MYH11 variant. Overall, the study presents useful insights for the field. While the rationale for most methodologies is well-explained, several aspects require clarification and improvements.

Major comments:

1. The results on pages 5-6 focus on the proband with the identified variant. However, this section lacks supporting data. Including some data on aortic root dilation or variant sequencing results would strengthen the findings beyond the text descriptions.

2. The identified VUS is common (0.6%) in the European population, making it relatively unlikely that it is a disease-causing VUS. Since the proband has pectus excavatum, could the aortic root dilation be associated with this condition rather than the VUS MYH11 p.Glu1893Asp variant? Is there any available information regarding potential cardiovascular phenotypes associated with this variant in the European population?

3. Figure 1: The figure legend title states that it is a homozygous mutation in the mice (Myh11E1892D/E1892D). But the sequencing in A on the right side clearly shows that it is heterozygous with the presence of a G and a C peak. This is confusing requiring clarification. The authors should include sequence data of both the heterozygous and the homozygous mice because they used both in the paper.

4. The cardiovascular phenotype of 13-month-old male and female mice was analyzed in Figure 1. The authors observed significant enlargement of the ascending aorta in female homozygous mutant mice compared to wild-type females, but this phenotype was not present in males. However, for further analyses in Figure 2A involving smooth muscle cell (SMC) contractility in aortae using myograph analysis, the authors utilized 3 male and only 1 female mice at 10 months of age. It is unclear why the authors combined sexes for this analysis, especially given the aortic dilation observed in females but not males in Figure 1 and then later the severe phenotype in male mice after TAC in Figure 4. This is especially confusing considering the authors provide a detailed discussion on sex-specific cardiovascular phenotypes (Pages 10-11). Moreover, the sex of the wt, het and homo mice used in Figure 2B at 6-month age is not specified. The rationale for changing the analysis time point to 10 and 6 months as opposed to the initial time point of 13 months is not explained.

5. Figures 3 and 4: It is not clear from the figure legend whether the histological data provided in 3A, 3C, and 4E is for male or female mice or both combined. If it is only for male mice, then the female mice data should be provided. If both sexes were combined, then given the sex-specific differences observed in Figures 1 and 4, analyses of the two sexes should be conducted separately for wild-type and Myh11E1892D/E1892D mice in 3A, 3C and 4E. The sample size for male and female mice in these figures is not indicated and should be provided.

6. Throughout the manuscript, it is unclear what statistical tests were used for each analysis. This is particularly important in cases where the sample size is n=2. The statistical methods should be clearly stated in the figure legends, and a minimum sample size of n=3 is necessary to ensure reliable conclusions. The sample size should be increased in experiments where n=2 or less for each sex per group.

7. The description of collagen quantification and elastic fiber fragmentation based on picrosirius-stained images is missing under the methods section and should be included.

Reviewer #3: In the enclosed manuscript, Zhou et al. have undertaken the important task of studying one of a growing number of genetic VUS identified in a typical clinical encounter with a patient with aortopathy. With the increasing use of genetic testing in patients with aortic aneurysm/dissection, these variants create clinical challenges for affected patients and family members.

Here, the authors investigate the effects of MYH11 p.Glu1892Asp variant on aortic function in a murine model. Interestingly, mice homozygous for the E1892D variant showed normal aortic diameters, protein expression of typical contractile markers, and response to vasoactive substances compared to wild-type controls. There was, in other words, no overt aortic phenotype in Myh11^E1892D/E1892D mice. Nevertheless, in a provoked central hypertension model with transverse aortic constriction, these mice showed a greater degree of aortic remodeling, elastin degradation, and aortic root diameter. Taken together, this reviewer interprets the E1892D variant as insufficient to promote aortopathy in isolation, but as a disease accelerant in the setting of a second insult (e.g., hypertension). The authors also report functional changes in left ventricular function (reduced), hypertrophy (increased), and enriched perivascular fibrosis in E1892D mice.

I commend the authors for a thorough investigation of the effect of this clinically identified VUS on aortic function in a mouse model. I have the following comments/questions.

1) Contractile function appears to be preserved in E1892D mice at baseline, however the histologic changes in TAC treated male mice appears to suggest a heightened "synthetic" phenotype in response to provocation. There are multiple reports of aortic smooth muscle cell phenotype change in aneurysm and hypertension/TAC models. Can the authors assess (either with histologic or PCR methodology) WT and E1892D aortas at baseline and with TAC to assess for recognized markers of SMC phenotype shift to determine if the E1892D variant somehow "primes" cells for progression to a deranged identity in response to stimulus

2) Like the authors, I find the changes in left ventricular function between E1892D mice and WT in response to TAC striking, however I find the mechanistic explanation for this finding and its relevance to human disease difficult to follow. Is this finding attributed to some change in aortic stiffness/resistance (beyond BP measurements), coronary vascular function (as evidence by remodeling in these vessels), or some other mechanism? Multiple plausible explanations are presented in the discussion, however none of these mechanisms are tested in the treated tissues.

3) Changes in aortic histology and diameter following TAC did not translate to aortic deaths in E1892D mice; do the authors propose that this is the result of short follow up time/constraints of the TAC model (which also necessarily produces a ventricular phenotype) or a suggestion of a fairly mild aortic phenotype in homozgous E1892D animals

Reviewer #4: In this manuscript, the authors describe the cardiovascular phenotype of a novel genetically modified mouse strain that models a variant of unknown significance in the MYH11 gene. While at baseline there are no significant differences between mutant and control mice (except for a mild ascending aortic dilation in aged female mutant mice), adding additional stress by TAC surgery did provoke a stronger phenotype in the MYH11 mutant mice. The strength of this study is that a new mouse model was generated to test the in vivo effects of the MYH11 variant on cardiovascular function, before and after surgically induced hemodynamic stress. Several issues remain to be addressed in order to support the relevance of the conclusions made by the authors.

It is not clear whether the proband was heterozygous for the MYH11 p.Glu1892Asp VUS, or whether it was present on both alleles. In the former case, how do the authors reconcile this with the fact that only the homozygous mutant mice appear to show a phenotype (based on the data in Fig. 2)? Were any aortic or cardiac measurements performed in heterozygous mice at baseline and/or after TAC? In addition, the E1892D variant is currently classified as benign or likely benign in Clinvar (as acknowledged by the authors), based on multiple submission highlighting that it is present in 0.5% of chromosomes in gnomAD and ExAC, represents a conservative amino acid change, and is predicted to be benign based on multiple in silico models. How would the authors propose to re-classify the variant based on the current data? Should it be considered as a variant leading to predisposition for aortic disease and/or cardiomyopathy in combination with other risk factors?

Which mechanism do the authors propose leads from the conservative substitution of Glu to Asp at position 1892 in MYH11 to the observed predisposition to aortic dilation and cardiac remodeling after pressure overload? Are there any in vitro studies that support an effect of this modification on protein function? The (sparse) data in Fig.2 suggests that the mutation does not appear to have an effect on protein expression – although this would need to be confirmed with more samples (see other comment below).

The studies referred to in line 66-67 made use of a fragment of the MYH11 promoter to drive Cre expression, which indeed was restricted to SMC. However, can the authors exclude that the endogenous MYH11 promoter, which might be affected by additional long-range enhancers or other genomic modulators, does not lead to leaky expression in other cell types at any point during development? This is important to check as one of the main hypotheses of the manuscript is that defects in vascular SMCs in the heart can contribute to pressure overload-induced cardiac hypertrophy and failure.

The manuscript lacks any information on how the Myh11 p.E1892D mouse line has been generated. Fig. 1 panel A seems to contain some markings relevant to the CRISPR strategy used, but they are not explained in the manuscript. Was the synonymous missense variant c.5679C>A introduced during CRISPR genome editing? Did the authors check for off-target effects of CRISPR/Cas editing?

Based on the role of MYH11 variants in cardiac rhythm (mentioned in the discussion, lines 268-271): did the authors perform any ECG measurements in the E1892D mutant mice? Was basal heart rate (as measured during echocardiography, but unfortunately not shown in the manuscript) different from wild-types?

The western blot data shown in Fig. 2 panel B only includes 2 samples for both the WT and mutant groups. This is not sufficient to be able to draw any meaningful conclusions, and needs to be repeated with larger numbers.

Finally, and importantly, a large part of the discussion focuses on many potential pathways that may be affected in the MYH11 mutant animals (IGF-1, PI3K/Akt/mTORC1, apoptosis, renin-angiotensin-aldosterone system, TGF-beta, inflammation, …). However, this is all very speculative and not based on any actual data collected in the current study. While it cannot be expected that the authors explore all pathways in detail in this study, the manuscript would strongly benefit from additional molecular data which might implicate any of these pathways in the observed phenotype. This could be achieved either by additional western blot data (including from animals that underwent the TAC procedure), QPCR, immunostaining, or potentially more unbiased methods including (single cell/nucleus) RNA-sequencing.

Minor comments:

- The rationale to test TAC in this model instead of Angiotensin II and/or L-NAME infusion, as was done in other models investigating MYH11 variant function, is not clear.

- The section on lines 205-210 on FBN1 variants does not seem to fit with the rest of the discussion – it should be made more clear what the point is the authors want to make here.

- Fig. S2 lacks the top of panels A and B

**Have all data underlying the figures and results presented in the manuscript been provided?**

Reviewer #1: Yes

Reviewer #2: **No: ** Details are described in the Comments for the Authors.

Reviewer #3: Yes

Reviewer #4: **No: ** Raw data are missing for all graphs shown in the manuscript. Particularly the echocardiographic data needs to be shown in more detail. The original western blots should also be shown in the supporting information.

PLOS authors have the option to publish the peer review history of their article (what does this mean? ). If published, this will include your full peer review and any attached files.

**Do you want your identity to be public for this peer review?** For information about this choice, including consent withdrawal, please see our Privacy Policy .

Reviewer #1: No

Reviewer #2: No

Reviewer #3: No

Reviewer #4: No

---

## [Decision Letter · Decision Letter 1]

PGENETICS-D-24-00923R1

MYH11 rare variant augments aortic growth and induces cardiac hypertrophy and heart failure with pressure overload

PLOS Genetics

Dear Dr. Milewicz,

Thank you for re-submitting your manuscript to PLOS Genetics. Your manuscript appears nearly ready for acceptance, however one reviewer did request some minor modifications of the text to provide more information to the reader. This request appears very reasonable and I do feel that the readers would benefit from working to clarify the two points identified by the reviewer. Therefore, we invite you to submit a revised version of the manuscript that addresses the points raised during the review process.

Please submit your revised manuscript within 30 days Jun 15 2025 11:59PM. If you will need more time than this to complete your revisions, please reply to this message or contact the journal office at plosgenetics@plos.org. Please include the following items when submitting your revised manuscript:

We look forward to receiving your revised manuscript.

Kind regards,

Chad S. Weldy, M.D., Ph.D.

Guest Editor

PLOS Genetics

Hua Tang

Section Editor

PLOS Genetics

Aimée Dudley

Editor-in-Chief

PLOS Genetics

Anne Goriely

Editor-in-Chief

PLOS Genetics

**Reviewers' comments:**

Reviewer's Responses to Questions

**Comments to the Authors:**

Reviewer #1: the authors addressed my minor concern

Reviewer #2: All critique points have been adequately addressed.

Reviewer #4: The authors have addressed most of my prior comments satisfactorily, but only two more points remain.

1) The most important point relates to my previous question #4, which I’m afraid the authors may have misunderstood. What I meant to say was that in the manuscript the authors use evidence from prior studies which make use of a fragment of the mouse Myh11 promoter to drive Cre expression as a basis to assume that the expression of Myh11 is restricted to SMC. Nevertheless, in the current manuscript the authors indeed performed CRISPR manipulation of the endogenous Myh11 gene. The expression of this gene is however driven by the full-length endogenous promoter (which could be influenced by distal elements and genome topology), and not only by the fragment used to demonstrate SMC-specific expression. My question therefore is whether there is conclusive evidence that the endogenous, genomic Myh11 promoter leads to the exact same expression pattern as the shorter fragment used to drive Cre expression in the prior studies cited by the authors. This is important as a key element of the conclusions put forth in this manuscript is that the observed phenotypes are exclusively SMC-dependent.

2) Thank you for clarifying that the proband is heterozygous for the MYH11 variant. It could be interesting to add a short line in the discussion on the point of using homozygous mutant mice for this study. The authors can indeed highlight that in a number of other mouse models of thoracic aortic disease a relevant phenotype can only be observed when the pathogenic variant or other mutation is biallelic. This does however raise some questions regarding the potential translation from mouse to human and vice versa: does the mouse model have more genetic redundancy? Or is the mouse aorta more resistant to heterozygous genetic defects?

**Have all data underlying the figures and results presented in the manuscript been provided?**

Reviewer #1: Yes

Reviewer #2: Yes

Reviewer #4: Yes

PLOS authors have the option to publish the peer review history of their article (what does this mean? ). If published, this will include your full peer review and any attached files.

**Do you want your identity to be public for this peer review?** For information about this choice, including consent withdrawal, please see our Privacy Policy .

Reviewer #1: No

Reviewer #2: No

Reviewer #4: No

**Figure resubmission:**
---

## [Editor Report · Decision Letter 2]

Dear Dr Milewicz,

We are pleased to inform you that your manuscript entitled "MYH11 rare variant augments aortic growth and induces cardiac hypertrophy and heart failure with pressure overload" has been editorially accepted for publication in PLOS Genetics. Congratulations!

Yours sincerely,

Chad S. Weldy, M.D., Ph.D.

Guest Editor

PLOS Genetics

Hua Tang

Section Editor

PLOS Genetics

Aimée Dudley

Editor-in-Chief

PLOS Genetics

Anne Goriely

Editor-in-Chief

PLOS Genetics

Comments from the reviewers (if applicable):

**Data Deposition**

http://datadryad.org/submit?journalID=pgenetics&manu=PGENETICS-D-24-00923R2

**Press Queries**

---

## [Editor Report · Acceptance letter]

PGENETICS-D-24-00923R2

MYH11 rare variant augments aortic growth and induces cardiac hypertrophy and heart failure with pressure overload

Dear Dr Milewicz,

We are pleased to inform you that your manuscript entitled "MYH11 rare variant augments aortic growth and induces cardiac hypertrophy and heart failure with pressure overload" has been formally accepted for publication in PLOS Genetics! Your manuscript is now with our production department and you will be notified of the publication date in due course.

With kind regards,

Lilla Horvath

PLOS Genetics

On behalf of:
